# Spatio-Temporal Evolution and Driving Mechanisms of Rural Residentials from the Perspective of the Human-Land Relationship: A Case Study from Luoyang, China

**Hua Wang** [1], **Yuxin Zhu** [1], **Wei Huang** [1], **Junru Yin** [1] and **Jiqiang Niu** [2,*]

[1] School of Computer and Communication Engineering, Zhengzhou University of Light Industry, Zhengzhou 450002, China; 2013016@zzuli.edu.cn (H.W.); zhuyuxin1010@163.com (Y.Z.); hnhw235@zzuli.edu.cn (W.H.); yinjr@zzuli.edu.cn (J.Y.)
[2] School of Geographical Sciences, Xinyang Normal University, Xinyang 464000, China
* Correspondence: niujiqiang@xynu.edu.cn

**Abstract:** Rural residential area is the core component of rural land, and the process of its transfer exists to guide the change of rural land use. Its evolution is closely related to population migration, so exploring the spatial and temporal evolution of the coupling type of human-land relationship and its driving mechanism is the core scientific basis for the optimal allocation of rural land resources. From the perspective of human-land relationship, this paper introduces spatio-temporal big data on the basis of 3S technology and constructs an elasticity coefficient model. The composite index method is also used to further explore the coupling types of rural human-land relations in the Luoyang region. At the same time, spatial autocorrelation and spatial autoregressive models were used to reveal the driving mechanism of the spatio-temporal evolution of rural human-land relationship types in the Luoyang region. The results showed that: (1) Against the background of the continuous reduction of the rural population in Luoyang, the scale of rural residential area first increases and then decreases, and spatially they continue to gather around the central city (2) At the township scale, the conflict between people and land tends to be moderated. The elasticity coefficient shows a zonal distribution pattern in the southwest of Luoyang region and the townships around the central city. (3) There is a non-negligible spatial correlation between rural human-land relations in Luoyang region. The initial scale of rural residentials, arable land area, rural resident population and industrial output value are positively correlated with the coordination of people-land relationship, and the slope, elevation and distance from the central city have an inverse effect on the harmony of the relationship between people and land.

**Keywords:** rural residential area; Luoyang region; spatial autocorrelation; man-earth relationship; spatio-temporal evolution; land resource allocation





## 1. Introduction

Against the backdrop of continuous urbanization, China's rural areas have entered a new phase of transformation and development [1]. This is primarily because the Party Central Committee has attached great importance to the construction of new socialist rural areas since the 18th Party Congress [2]. However, unlike the urban planning system, rural land resources have not been compiled in a scientific and reasonable manner [3]. Moreover, with the continuous promotion of urbanization and modernization in China, the incongruity between urban and rural development has been increasingly highlighted. The relationship between humans and land has tended to be unbalanced, gradually approaching the state of "land division, urban-rural division, and human-land separation" [4]. According to the land survey results published by the Natural Resources Bureau, urban construction land in China in 2015 was less than half that of rural residential land; the per capita residential land area of the rural household population was as high as 218.32 $m^2$ [5]. This illustrates

the severity of the existing imbalance in the human-land relationship centered on the rural population. Systematic analysis of the spatio-temporal evolution of rural residentials and its driving mechanisms would provide a theoretical basis for this phenomenon and the technical support required for realizing efficient and intensive use of rural land [6–8].

Recent research has focused on the changes in land use brought about by the evolution of rural residential areas. These studies analyze the relationship between humans and land, and the majority of scholars have drawn on the relevant research results of the Land-use/Land-cover change (LUCC) evolution simulation as a theoretical basis [9]. A brief literature review revealed that studies on the evolution of spatial and temporal land-use patterns by foreign scholars focus primarily on the analysis of modeling methods and model accuracy [10,11], factors influencing layout evolution [12–15], and driving mechanisms [10,16–18]. Similarly, most of the studies conducted in China are devoted to analyzing the characteristics of landscape evolution [19,20], finishing potential [21], and the layout optimization schemes of rural residential sites [22,23], with limited research on the spatio-temporal evolution of human-land relationships. These studies focus primarily on the macroscopic scale, covering a large area and with large spatial units [24], with fewer studies conducted at the microscopic scale, which is more difficult for spatial heterogeneity research. From the geographic point of view, determining the comprehensive potential of rural residentials mainly involves analyzing the suitability of natural and socio-economic factors, neglects the spatial correlation among rural residentials [25], and ignores the role of neighborhood influencing factors [26,27]. In terms of research methods, the majority of studies focus on analyzing the driving mechanisms and trends in the evolution of rural residentials using geographic information spatial technology [28] or by constructing a single simulation model [29]. However, such methods cannot incorporate the dynamic tracking and simulation of the evolution of spatio-temporal patterns of human-land relations. To address this gap, this paper introduces spatio-temporal big data based on "3S" technology (Geographic Information Systems, Remote Sensing, and Geographic Positioning Systems) from the perspective of human-land relationships. It aims to dynamically analyze the characteristics of temporal changes and the spatial divergence patterns of rural residential land use and rural population in Luoyang region during 2009–2019, using townships as spatial units. An elasticity coefficient model is constructed and combined with the comprehensive index method to further explore the coupling types of rural human-land relations. In addition, spatial autocorrelation and spatial autoregressive models are used to determine the driving mechanisms of the spatial and temporal evolution of these relations. Consequently, it provides a more efficient digital guidance scheme for rural land resource management projects in China as a basis for achieving more scientific, effective, and optimal allocation of such resources.

## 2. Overview of the Study Area and Research Methodology

### 2.1. Overview of the Study Area

The study area, Luoyang City, is located in the western part of Henan Province in central China. It is considered the birthplace of Chinese civilization and is the deputy center city of the Central Plains City Cluster. The region straddles the north and south banks of the middle and lower reaches of the Yellow River, extending from 112°16′ E to 112°37′ E and 34°32′ N to 34°45′ N. China's administrative divisions are divided into provincial-municipal-county-township levels, so the region of Luoyang contains many townships, and the villages are located in these townships. Luoyang region has 7 counties and 7 districts under its jurisdiction, the total area of the it is 15,230 km$^2$ and rural residentials account for 32.18% of this area. At the end of 2019, the total population of Luoyang region was 7,170,200, of which the rural population accounted for 43%. Luoyang region has a diversified topographic structure, consisting primarily of mountains, hills, and plains. The mountains are mainly in the southwest, while downtown from Luoyang are the plains. The middle transition zone is hilly. In addition, Luoyang region is very rich in water resources,

mineral resources, tourism resources, and so on. The location and map of the city are shown in Figure 1.

**Figure 1.** Location and map of Luoyang region.

### 2.2. Data Source and Processing

### 2.2.1. Data Source

The primary source of data for this study were the three land-use change survey databases of Luoyang region in 2009, 2014, and 2019. Table 1 lists these sources.

**Table 1.** Description of data sources.

| No. | Data | Description of Data Source | Department of Data Source |
|---|---|---|---|
| 1 | Land use data and traffic and water conservancy planning data of Luoyang region in 2009, 2014 and 2019 | The second National Land Survey Project in 2009, the Land Change Project in Henan Province in 2014, and the third National Land Survey project in 2019 | Natural Resources Planning Bureau of Henan Province Land bureau of Henan Province |
| 2 | Original data of driving factors such as elevation, population and industrial output value of each township in Luoyang region in 2019 | Evaluation Results of Cultivated Land Quality Renewal and County Statistical Yearbook (2019) | Agriculture Bureau |
| 3 | Town bound boundaries | Industrial and tourism development planning | Development and Reform Commission |

### 2.2.2. Data Processing

Data preprocessing included two modules. The first is data correction and cleaning. Field surveys were conducted by randomly selecting dynamic patches for repeated interpretation and analysis to evaluate the classification accuracy, with an overall accuracy above 80%. The geographic boundaries of the administrative divisions of Luoyang region have been adjusted several times over the past ten years. For example, in 2009, there were more than 300,000 land use patches in Luoyang region, and more than 110,000 rural residential sites. Therefore, the towns and cities belonging to the central city were excluded, and only the data for the counties were retained. The rural residential sites in 2009 and 2019 are shown in Figure 2.

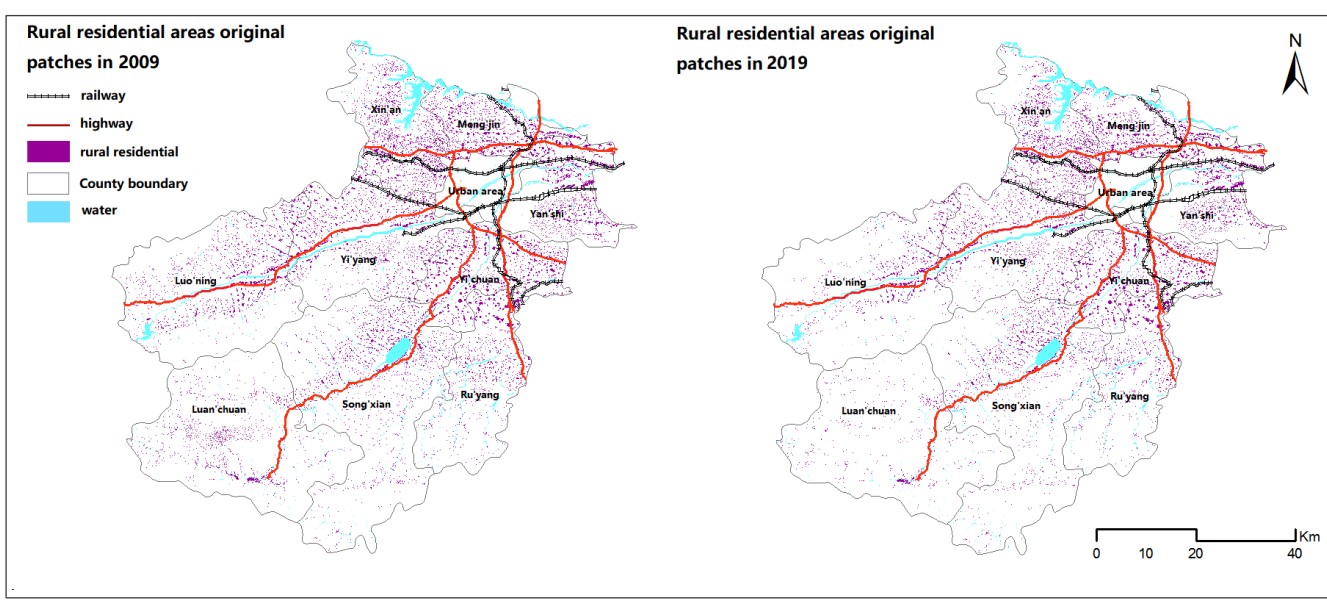

**Figure 2.** Rural residential areas—original patches in Luoyang region.

Second, it was ensured that the spatial extent of each type of data was consistent for each period. In this study, we used the Raster Calculator tool to extract the spatial extent information that overlaps completely after overlaying each layer and then used this layer as a mask to extract spatial land use data and the various driving factors for each phase. Then, reclassification (Reclass) was performed to obtain seven land use categories: arable land, forest land, water area, urban land, rural residential, other construction lands, and land used for transportation. Since gardens and grassland covered a small area, these categories were combined with forest land as forest and garden land. The spatial quantification and visualization of some of these drivers are shown in Figure 3.

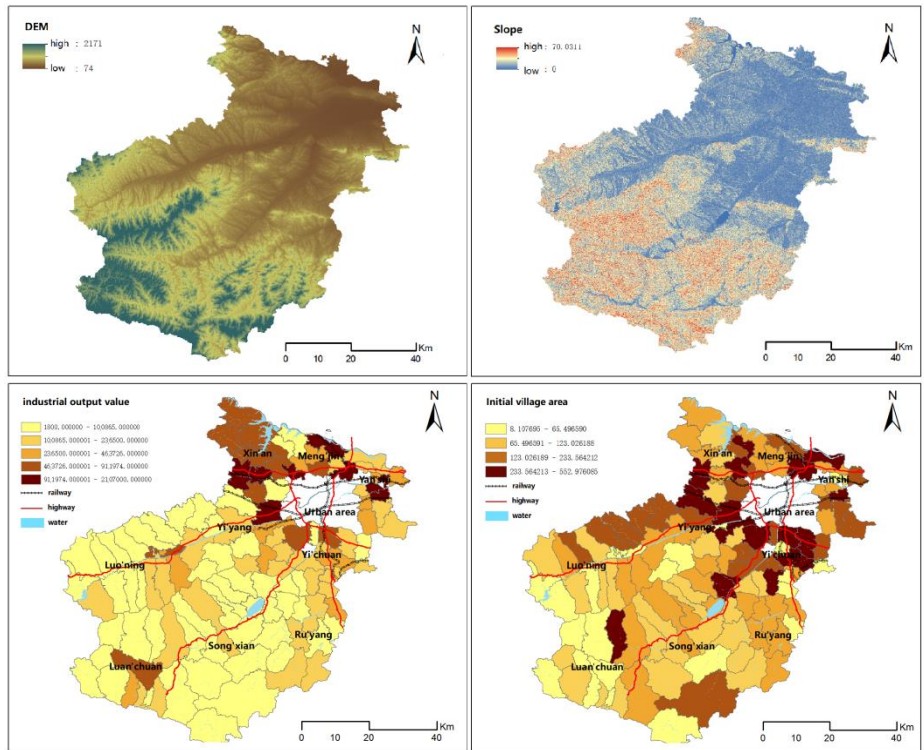

**Figure 3.** Spatial distribution of some driving factors.

## 3. Research Method

### 3.1. Elastic Coefficient

The scale and number of rural residentials are influenced by various natural and social factors, among which one of the most important is the change in rural population. In view of the gradual weakening of the difference between urban and rural household registration in China, this study uses the total rural resident population to determine this change. The People Net Rate (*PNR*) is used to represent the rate of change of the rural population. This quantity is calculated as shown in Equation (1). The land net rate (*LNR*) represents the rate of change of rural residential land area during the research period and is calculated as shown in Equation (2).

$$LNR_i = \frac{D_{(1+t)i} - D_{1i}}{D_{1i}} \times 100\% \tag{1}$$

$$PNR_i = \frac{P_{(1+t)i} - P_{1i}}{P_{1i}} \times 100\% \tag{2}$$

In the equations, $LNR_i$ and $PNR_i$ represent the net rate of change of rural residential land area and the rate of change of rural population, respectively, for unit$_i$. The variables $D_{1i}$ and $D_{(1+t)i}$ represent the rural residential land area of the first research unit at the beginning and end of the research period, respectively. $P_{1i}$ and $P_{(1+t)i}$ represent the rural population of the *i*th research unit at the beginning and end of the research period, respectively. To further analyze the relationship between rural population migration and the change in the scale of rural residential land, an elastic coefficient model was constructed. This model analyzes the coupling type of the rural man-land relationship in Luoyang and the elastic coefficient (*EC*) is obtained as shown in Equation (3).

$$EC_i = LNR_i / PNR_i \tag{3}$$

### 3.2. Spatial Correlation Analysis

The first law of geography has always been the theoretical basis for spatial autocorrelation analysis. As a continuous space, rural residential land also satisfies this law, since spatial correlation or similarity exist. Global spatial autocorrelation can be performed to calculate the degree of dependence of a research unit on the total spatial scope. In contrast, local spatial autocorrelation analysis focuses on the spatial correlation between a specific unit and its immediate neighbors. Yu Huimin and other scholars describe these principles and the equations used for calculation [30].

### 3.3. Types of Coupling Relationship between Rural Residential Areas and Rural Population Change

According to the geological change principle of man-land synchronization, the changes in trends of rural residential land scale and rural population should be the same. However, in practice, there are a large number of residentials in many towns and villages, which lead to opposing trends. Therefore, to further explore the interactions and evolution of this relationship, the corresponding coordination degree [31] is defined in this paper based on whether land use is intensive or not, and constructs the rural man-land coupling relationship model (Table 2). In the case of complex increasing and decreasing trends of the two quantities, the coupling relationship is divided into the following eight types.

**Table 2.** Spatial distribution of some driving factors.

| Type | LNR | PNR | EC | Coordination or Not | |
|------|-----|-----|-----|--------------------|---|
| A | + | + | $[1, +\infty]$ | not | |
| B | + | + | $[0, 1)$ | coordinate | |
| C | − | + | $[−1, 0)$ | coordinate | |
| D | − | + | $(−\infty, −1]$ | coordinate | |
| E | − | − | $[1, +\infty]$ | coordinate | |
| F | − | − | $[0, 1)$ | not | |
| G | + | − | $[−1, 0)$ | not | |
| H | + | − | $(−\infty, −1]$ | not | |

*3.4. Spatial Autoregression*

Traditional statistical methods that are used to analyze the driving factors affecting the man-land relationship rely on the uniform and independent distribution of data in the research area. An example of such a method is the classical linear regression model, which makes this assumption [32]. However, in reality, spatial autocorrelation exists in geographic data. The spatial autoregressive model (including the spatial lag model and spatial error model) can be modified to include spatial dependence in the regression equation. Therefore, the spatial autoregressive model is selected to analyze the driving mechanism of the man-land relationship in the rural areas of Luoyang region. The equations for the model are as follows:

$$Y = \rho W_1 + \beta X + \mu$$
$$\mu = \lambda W_2 + \varepsilon \tag{4}$$

where, $Y$ is the dependent variable; $X$ is the explanatory variable; $\beta$ represents the spatial regression coefficient of explanatory variables. $\mu$ is the error term; $\varepsilon$ is white noise; $W_1$ is the spatial weight matrix reflecting the spatial trend of the dependent variable, and $W_2$ is the spatial weight matrix reflecting the spatial trend of the residual. $\rho$ is the coefficient of the spatial lag term, $\lambda$ is the spatial error coefficient, the range of the two is $[0, 1]$ and. the closer the two are to 1, the more similar are the values of the dependent variable or explanatory variable in the adjacent region.

## 4. Results Analysis

*4.1. Spatial Autoregression*

From the perspective of temporal characteristics, the overall rural population of Luoyang region showed a continuous downward trend from 2009 to 2019, with a total decrease of 458,574 people. In 2014, the area covered by rural residentials in Luoyang region increased to 868.44 km$^2$ and the relationship between man and land appeared to be severely imbalanced. In 2019, the scale of rural residential land in Luoyang region decreased to 668.54 km$^2$. During this period, the area of land under construction in Luoyang town increased by 191.08 km$^2$. This was due to the migration of a large number of people from rural to urban areas during the urbanization process as well as the separation of people and land in rural areas, resulting in a "double increase" of urban construction land and rural residential land area.

Using the comprehensive index method, the change in the rural resident population and the change in the rural residential land scale can be divided into active increasing type (PNR&LNR$\in[1, +\infty)$), stable increasing type (PNR&LNR$\in [0, 1]$), active decreasing type (PNR&LNR$\in(−\infty, −1]$), and stable reduction (PNR&LNR$\in[−1, 0]$). The spatial distribution of each type is shown in Figure 4.

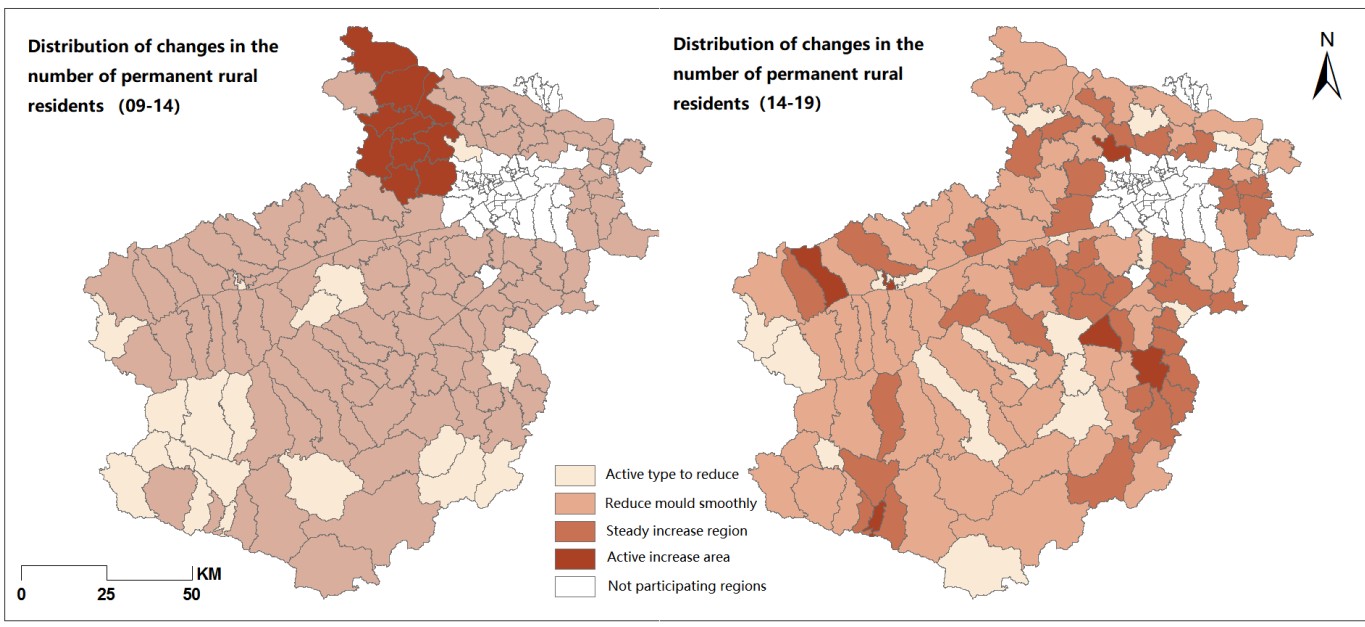

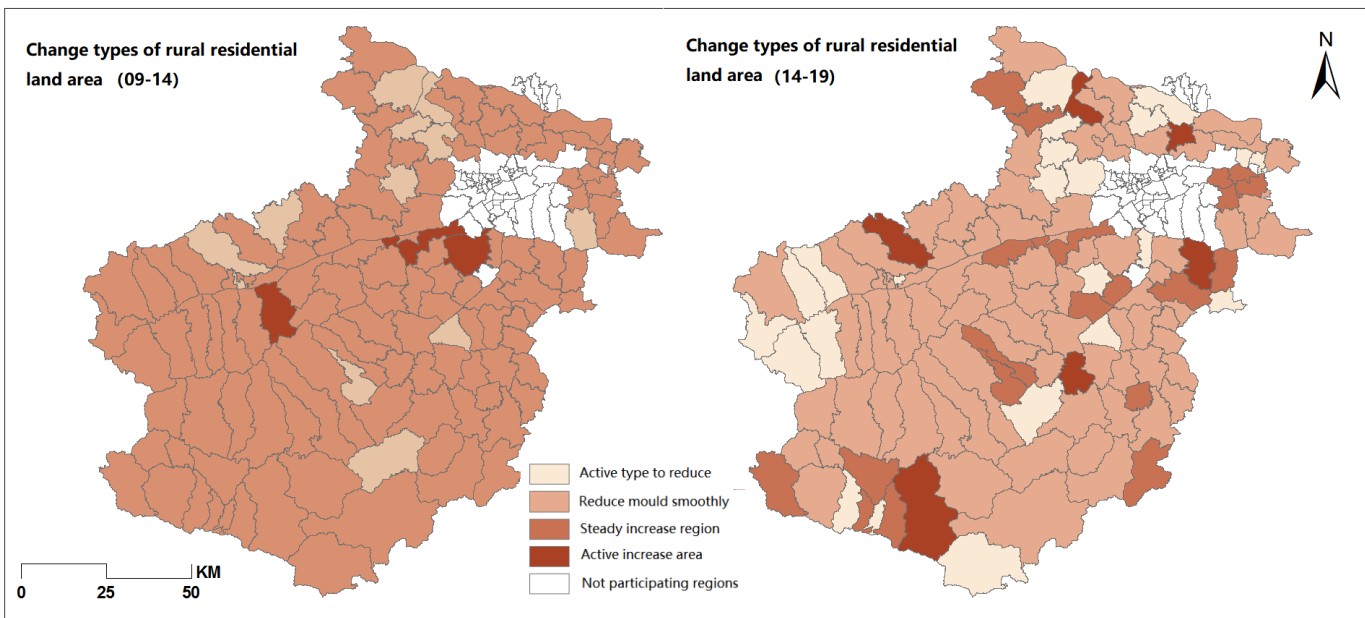

**Figure 4.** Spatial distribution of the change types of rural population and rural residential land scale.

In terms of spatial distribution, between 2009 and 2014, 84.6% of the city's townships have seen a steady decline in the number of rural people. However, in Xin'an County, the rural population in the majority of townships was actively increasing, apart from Caochun Township. This was because 14 village, including Shijing in Xin'an County, were formally abolished and established as townships between 2009 and 2011. The implementation of the township-run village system resulted in a corresponding increase in population size and regional area. During the same period, the scale of rural settlement land in 91.13% of townships in Luoyang region continued to increase, and the phenomenon of people leaving the land and separating from the land was serious. From 2014 to 2019, the rural population of 57.3% of towns and villages in Luoyang region exhibited a decreasing trend and the rural population reflux phenomenon was observed in the villages and towns surrounding the city center and to the east of Yi'chuan and Ru'yang County. During the same period, 79.03% of the townships in Luoyang region showed a decreasing trend in the size of rural settlements, while the types of changes in rural land tended to be consistent.

The dynamic change in rural residential land is not only reflected in the increase and decrease of spatial distribution but also in the mutual transformations with other land types. From 2009 to 2014, a total of 105.19 km$^2$ of land was converted into rural residential land, of which 77% was converted into farmland, forests, gardens, and grasslands. The main outflow direction was land used for farmland water conservancy facilities, grasslands, and forest park land, with a total outflow area of 32.6 km$^2$. From Figure 5, it can be observed that, during this period, the new rural residential sites in Luoyang region were spatially concentrated around the urban areas and the townships near the main roads. It is evident that most of them were converted from arable land, thus indicating that rural land resources were not effectively utilized or managed and that the occupied areas could have been more efficiently used for other purposes.

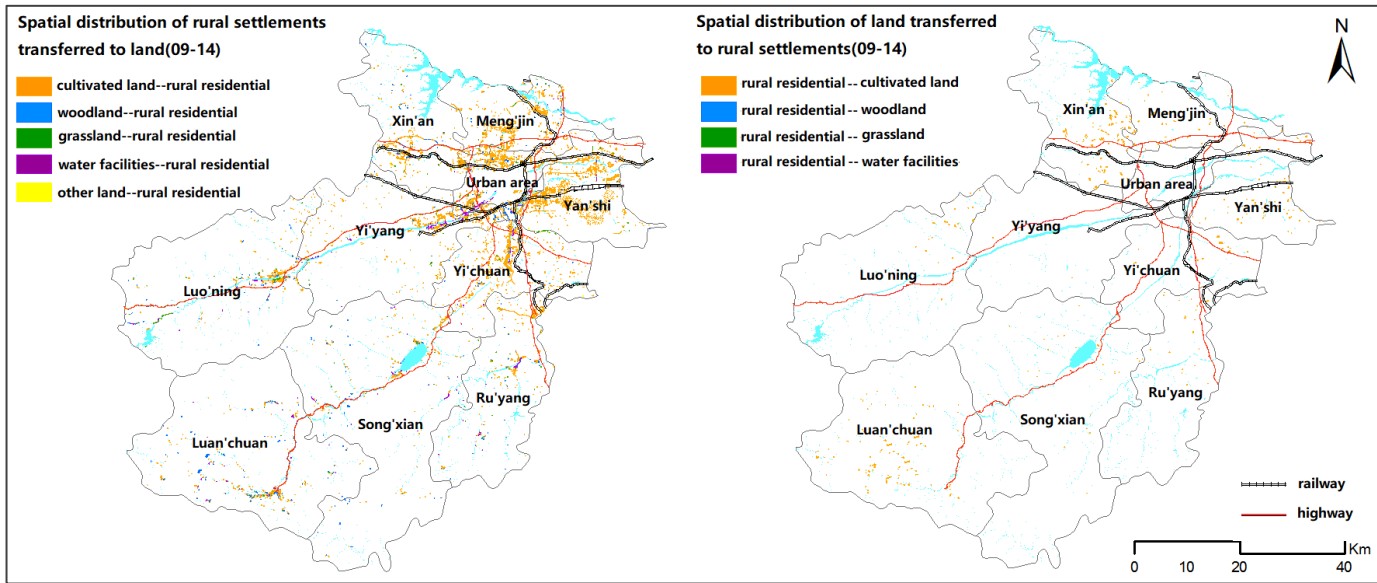

**Figure 5.** Spatial Distribution of Land Types Transferred to and from Rural residentials in Luoyang region (2009–2014).

The transfer out of rural residentials in Luoyang region during 2014–2019 was much larger than the transfer in. The sources of new rural residentials transferred during this period were still arable land, forest and garden land, and grassland. A total of 50.44 km$^2$ was transferred in, which was 31.26 km$^2$ less than the transferred area in the previous period, and only accounted for 25% of the transferred area. From Figure 6, it can be observed that most of the transitions between various land types and rural residentials during this period are concentrated in the southwestern counties, with their distribution concentrated along rivers and roads. When compared with the previous five years, the biggest difference is that arable land and other types of land have changed from the source of transfer in to rural residentials to the object of transfer out. This indicates that the unreasonable occupation of land resources by rural residentials has been significantly reduced and that the balance of natural resource occupation has achieved remarkable results.

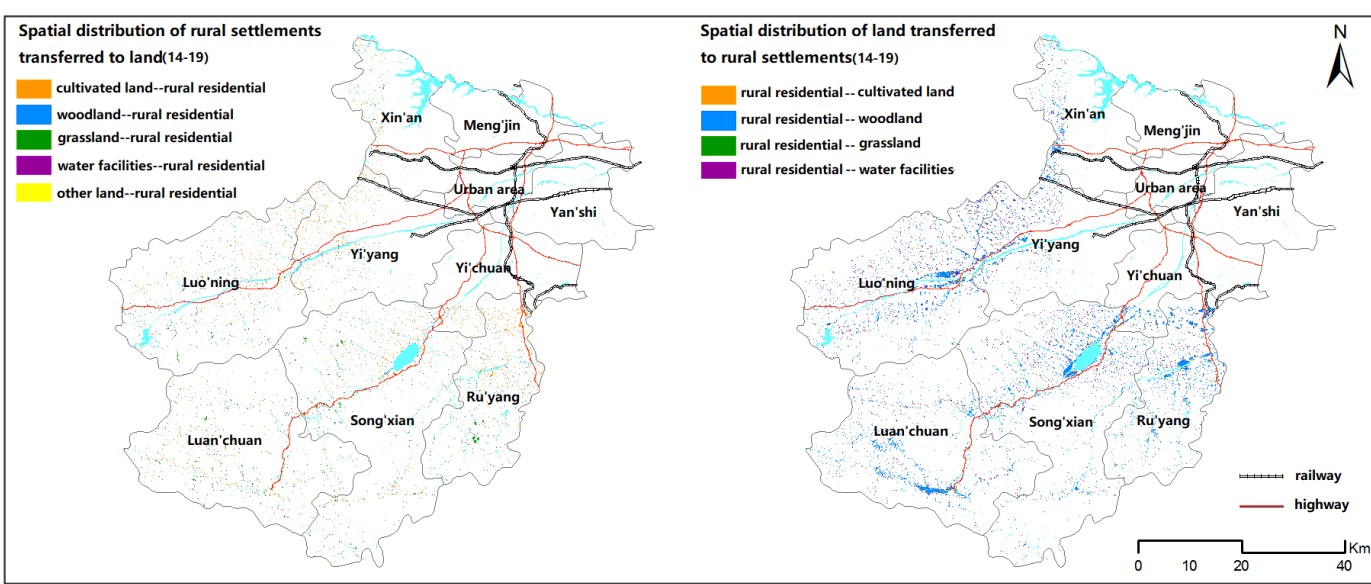

**Figure 6.** Spatial Distribution of Land Types Transferred to and from Rural residentials in Luoyang region (2014–2019).

*4.2. Spatio-Temporal Evolution Characteristics of Man-Earth Coupling Relationship Types Based on EC*

Because of the obvious spatial heterogeneity of human-land relations with townships as the spatial unit, the types of human-land coupling in Luoyang are classified into four major categories in this study. This is based on the criteria for classifying the types of human-land coupling shown in Table 2—both people and land increase, both people and land decrease, people decrease and land increases, and people increase and land decreases. Based on the calculation results of LNR and PNR, these four types of human-land coupling relationship types were subdivided according to the coordination. Figure 7 shows the visualization results of the spatial distribution of the subdivided types.

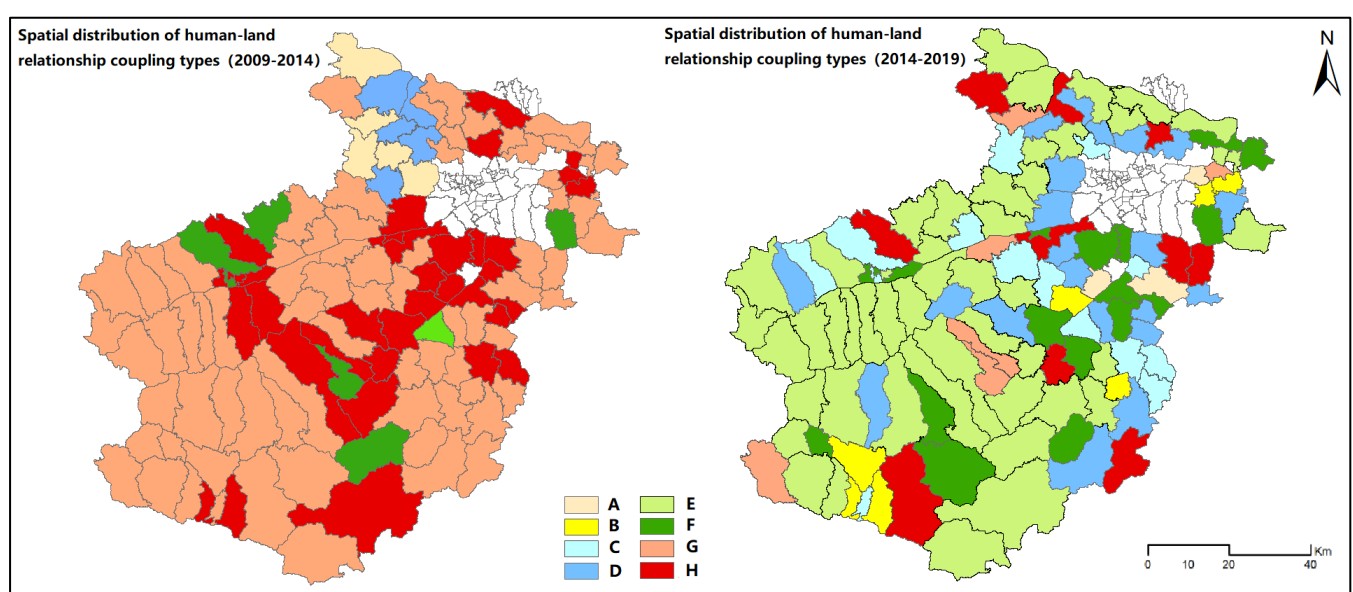

**Figure 7.** Spatial distribution of human-land relationship coupling types.

From Figure 7, it can be observed that the type of man-land coupling relationship in rural areas of Luoyang region from 2009 to 2014 was mainly uncoordinated with "man-land decrease and land-increase", and 33 townships were in a state of serious imbalance (red

area). From 2014 to 2019, the coordinated type of "simultaneous reduction of people and land" became the dominant type, with a total of 72 towns. Most of these were converted from the type of "reduction of people and increase of land" in the previous stage, such as Xiaolangdi Town, Huimeng Town, Yanzhen Town, Yiyang County, and Gaocun Town. However, most of the towns around the urban area, including Yi'chuan county and Ru'yang County, have changed into the "human-land coupling" type. This shows that, during this period, in most of the towns and villages the human-land relationship had transformed from uncoordinated development in the direction of more coordinated human-land development. During this period, the rural population in these townships has returned and the occupation of land resources by rural settlements has improved greatly.

By comparison, it can be seen that most of the townships have shifted from the direction of uncoordinated development of human-land relationship to coordinated development of human-land. This is closely related to the continuous promotion of integrated urban-rural sustainable development and optimal land space allocation in China. In order to further study its evolution pattern, this paper extracts the areas where the type transformation occurred for spatial distribution visualization (as shown in Figure 8). The evolution of the spatial layout of human-land relations can reflect the level of economic and social development in the region very intuitively. Therefore, we took a typical sample from the townships where type switching occurred for in-depth analysis.

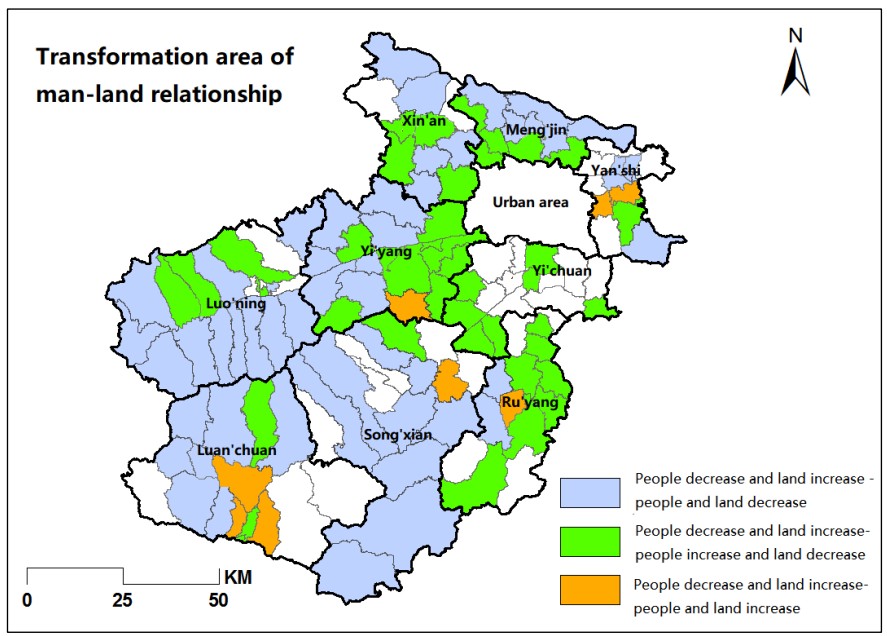

**Figure 8.** Distribution of human-land relationship coupling type transformation.

### 4.3. Typical Transect Analysisc

Among the townships in Luoyang region where the coupling between humans and land changed significantly during the two periods considered, those with spatially adjacent but different socio-economic conditions and topography were selected as typical sample zones. This was done based on the economic class, utilization level, and topography of each township. Then, we selected the north-south transect of the city to better analyze the counties and districts that transitioned from human-reduced land increase to human-increased land decrease, including Xin'an County to the north of Luoyang region center.

#### 4.3.1. Population Decrease and Land Increase—Population Increase and Land Decrease

From 2009 to 2019, the type of human-land coupling relationship changed in the townships in the eastern boundary of Luoyang region center from north to south, from Mengjin County to Xin'an County to Ruyang County. The spatial distribution of this sample zone is shown in Figure 9. The dysfunctional "people decreasing land increasing" type

gradually transitioned to the harmonious, coordinated "people increasing land decreasing" type. The reasons for this shift in the sample zone can be summarized in two ways:

1. Rural revitalization strategy to promote new rural construction: 14 villages in Mengjin County were selected for the pilot project of beautiful countryside construction in 2013. Consequently, the living environment and quality of rural life have continuously improved. In addition, in recent years, Ru'yang County's characteristic superior agricultural output value and per mu benefit of characteristic planting have increased significantly, while public service facilities and living environment have also improved.

2. Rapid development of the tourism industry: in recent years, Luoyang region has been vigorously developing rural tourism projects. These not only promote the employment of poor people and broaden the channels of income generation but also thoroughly organize and plan the utilization of rural land resources. Mengjin County, Luanchuan County, and Song County are representative counties that have developed a greater number of scenic patches and consistent policy support has attracted various groups to join this industry. This is also the main reason why the rural population in the typical sample counties has returned in the past five years, while the land size of rural residentials has decreased, thus resulting in the "increase in people and decrease in land".

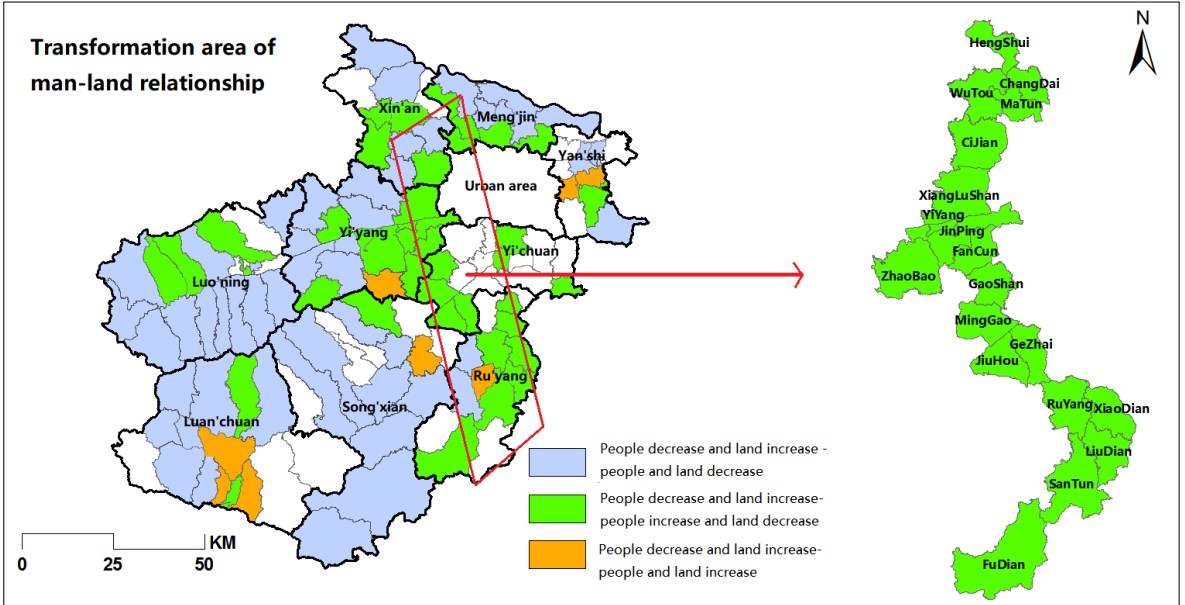

**Figure 9.** The location of the typical sample area of people decrease and land increase—people increase and land decrease.

### 4.3.2. Population Decrease and Land Increase—Population and Land Decrease Area

In Luo'ning County and Song County, the coupling relationship between people and land is subtractive. The spatial distribution of this sample zone is shown in Figure 10. The main reason for the transformation of this relationship is the efficient utilization of natural resources and the optimal allocation of land resources. These are both typical poor counties in mountainous areas, which is why the rural population is constantly migrating in the early stage of urbanization. However, the land left in rural areas for housing, breeding, or animal husbandry has not been properly handled, resulting in an imbalance of "population reduction and land increase". In recent years, driven by the national poverty alleviation strategy, the government has not only developed characteristic tourism and industrial parks from scratch but also established large-scale wind, water conservancy, and solar power generation projects. These have transformed the rural power grids in poor villages while making efficient and intensive use of land resources. This series of measures have resulted

in intensive rectification and redistribution of the unused residential land left in rural areas and also changed the relationship between man and land from the uncoordinated development state to the coordinated development type of "simultaneous reduction of man and land".

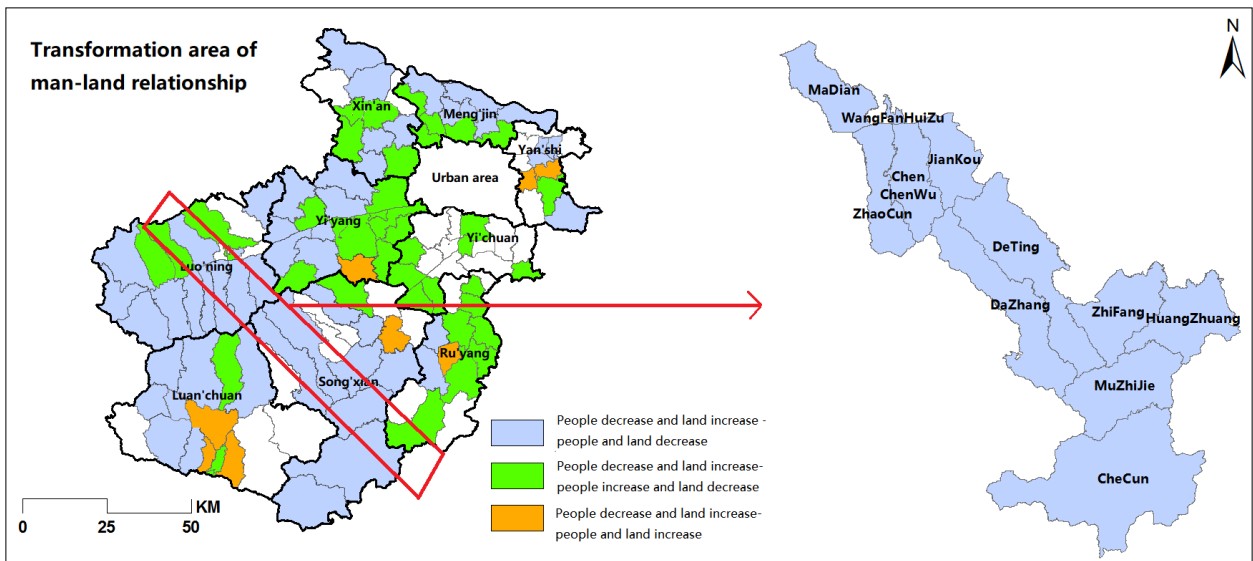

**Figure 10.** The location map of the typical sample area of people decrease and land increase—people and land decrease.

*4.4. Analysis of Driving Mechanism of Spatio-Temporal Evolution of Man-Land Relationship*

4.4.1. Spatial Correlation Analysis

Global and local spatial autocorrelation analysis can verify whether the elastic coefficient shows correlation and aggregation in spatial distribution. In this paper, by calculating Global Moran's I value of the elastic coefficient of towns in Luoyang region, the Moran scatter diagram is obtained as shown in Figure 11. The global Moran's I index for ECs for the two periods 2009–2014 and 2014–2019 were 0.2139 and 0.2633, respectively, and both passed the significance test. These results confirmed the spatial correlation of man-earth coupling relationship types between regions, i.e., the coordination of the man-earth relationship in the target region is affected by the neighborhood.

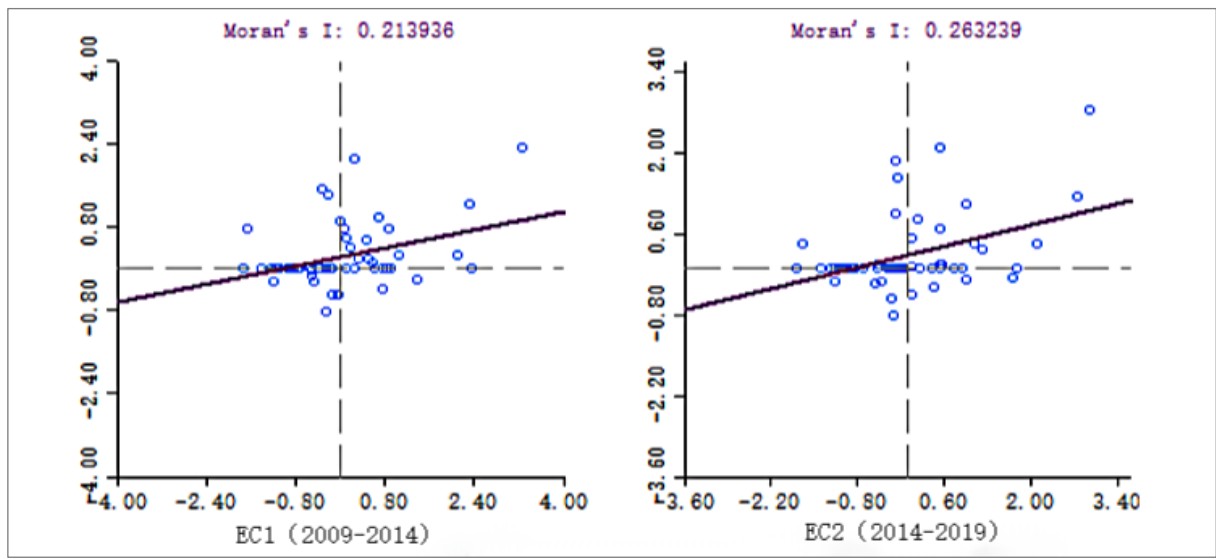

**Figure 11.** Scatter plot of the global Moran index of the 2−period elasticity coefficient.

The local spatial autocorrelation results are shown in Figure 12, where high-high type and low-low type indicate a positive spatial correlation between the man-land relationship type of the target township and that of the neighborhood, and that the evolution trend is consistent. Low-high type and high-low type indicate that the man-land relationship type of the target town is negatively correlated with that of its neighbors in space, i.e., the man-land relationship type of the target town is opposite to that of the surrounding towns. High-low type and low-high type are spatially similar to "convex ground" and "concave ground". According to the spatial polarization theory, the target area will eventually be assimilated by the surrounding area and then transition to either high-high or low-low type. Therefore, the spatial correlation should not be ignored in the process of land resource planning.

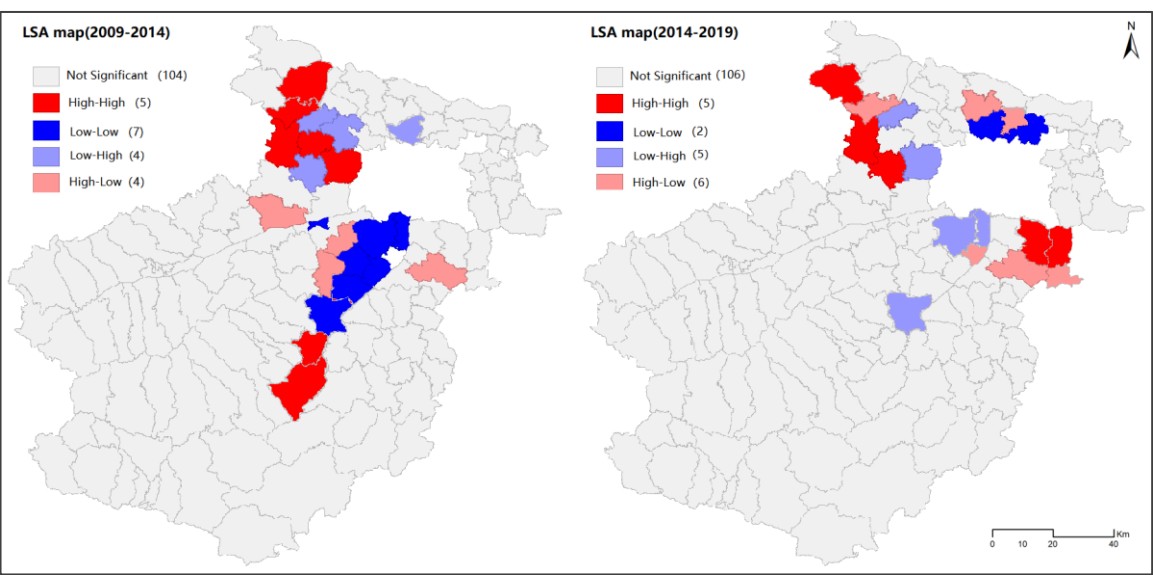

**Figure 12.** 2 period elasticity coefficient local spatial correlation aggregation map.

Moreover, it can be seen from Figure 12 that the spatial agglomeration of the man-land relationship in rural areas of Luoyang region is not obvious. Most of the high-value areas are concentrated in towns close to the city, while the low-value areas are distributed in towns with fewer people and more land. For example, during 2009–2014, the high-high type of relationship was concentrated at a distance from the city's new county town, north town, stone temple town, town and magnetic iron jian town. This is due to the fact that, after Xin'an County completed the abolition of townships in 2011, the government centralized construction and unified management, the economic development between townships drove each other and the level of public facilities remained basically the same, so the evolution trend of the relationship between people and land between townships is the same, and it belongs to the type of people and land increasing together. Therefore, the trend of evolution of the man-land relationship between villages and towns was consistent and could be classified as the man-land co-growth type. The low-low type of relationship was concentrated in Yi'chuan County, from Licun Town in the north to Tianhu Town in the south. Like Xin'an County, Yi'chuan County also implemented unified government management after the completion of township construction; however, because it is located in the hills, the overall economic level is relatively backward. Therefore, most villages and towns belong to the type of population reduction and land increase. Low-high and high-low towns with negative spatial correlation were distributed sporadically, accounting for 4.84% of the total number of towns. The ECs of 83.9% of townships had no significant spatial correlation.

4.4.2. Selection of Driving Factors

Since the changes in the human-land relationship are usually influenced by both environmental and socio-economic factors, ten indicators were selected that may potentially affect this relationship in Luoyang region. These included variables such as elevation, slope, number of total permanent rural population, urbanization rate, industrial output value, per capita income, distance from cities, distance from roads, arable land area, and initial size of the village. Bivariate spatial autocorrelation analysis was performed for the EC of the second phase (2014–2019) and the results are shown in Figure 13. The values of the Moran's I index show that there are obvious spatial correlations; therefore, the influence of the neighborhood on the type of man-land relationship in the target region cannot be ignored.

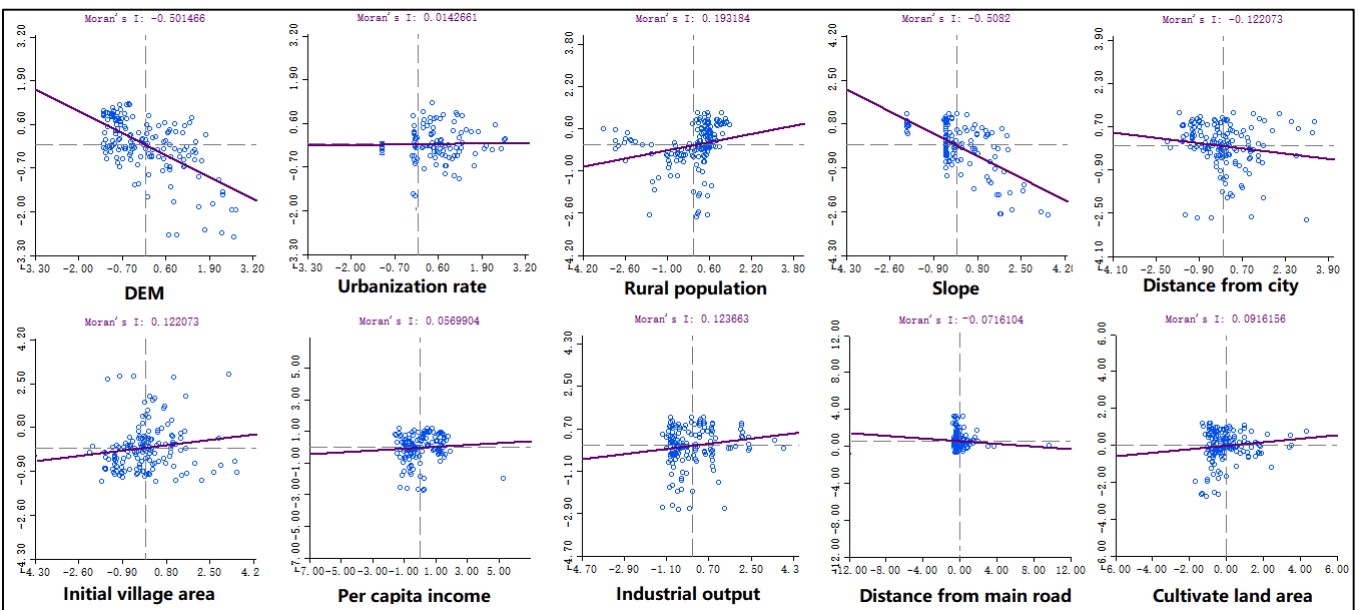

**Figure 13.** Bivariate spatial autocorrelation Moran index values.

4.4.3. Driving Mechanism Modeling of Spatio-Temporal Evolution of Man-Land Relationship

To further analyze the driving factors and the degree of their impact on the coordination of the man-land relationship, a spatial regression model was constructed, with each influencing factor as an independent variable and the EC as the dependent variable. Classical linear regression analysis using the method of Ordinary Least Squares (OLS) was performed before spatial autoregression analysis. Here, if other conditions remain unchanged, the regression coefficient represents the extent to which the dependent variable is affected by a change in a particular independent variable. T-statistics and *p*-values were used to test the significance of the influence of independent variables on dependent variables and the results are shown in Table 3.

According to the positive and negative coefficients shown in Table 3, it can be seen that the coordination of the man-land relationship is positively correlated with the permanent rural population, industrial output value, initial village area, urbanization rate, and cultivated land area. It is negatively correlated with slope, elevation, and distance from cities and roads. In addition, the parameters listed in the OLS model results as shown in Table 4 are used to select the spatial autoregressive model that is most consistent with the real-world scenario.

**Table 3.** OLS model calculation results (2019).

| Impact Factors | Coefficient | Standard Error | t-Statistics | *p*-Value |
|---|---|---|---|---|
| constant | 0.788471 | 0.043790 | 18.0056 | <0.0001 |
| DEM/m | −0.030123 | 0.003780 | 1.53636 | 0.12702 |
| Slope/° | −0.037962 | 0.004216 | −9.00427 | <0.0001 |
| Rural population/per | 0.000862 | 0.000982 | −0.60782 | 0.25148 |
| Industrial output/million | 5.16E-08 | 2.56E-08 | 2.01904 | 0.04566 |
| Initial village area/Km$^2$ | 0.00132278 | 0.000814639 | 1.62377 | 0.10648 |
| Distance from city/Km | −5.89531e-006 | 5.81535e-006 | −1.01375 | 0.31230 |
| Cultivated land area/Km$^2$ | 0.0012688 | 0.000182707 | 6.94446 | <0.0001 |
| Urbanization rate/% | 0.00241696 | 0.00527052 | 0.458582 | 0.64737 |
| Distance from main road/Km | −7.89347e-005 | 4.42925e-005 | −1.78212 | 0.07728 |
| Per capita income/yuan | 8.31935e-007 | 1.8275e-006 | 0.455231 | 0.64977 |

**Table 4.** Spatial correlation test.

| TEST | MI/DF | VALUE | PROB |
|---|---|---|---|
| Moran's I (error) | 0.159 | 3.1076 | 0.00189 |
| Lagrange Multiplier (lag) | 1 | 13.2549 | 0.00027 |
| Robust LM (lag) | 1 | 6.3343 | 0.01184 |
| Lagrange Multiplier (error) | 1 | 6.9736 | 0.00827 |
| Robust LM (error) | 1 | 0.053 | 0.81791 |

As can be seen from Table 4, Moran's I for the residual of the classical linear regression model is 0.159 and is significant, thus indicating that the residual has spatial autocorrelation. This result justifies the need to introduce the spatial autoregression model. Anselin proposed a standard for selecting a spatial autoregressive model. Firstly, the significance of lag and error in Lagrange Multipliers (LM) are determined. If either is significant, the corresponding spatial autoregressive model is used. If they are both significant, the robust forms of the multiplicative numerical LMs, i.e., Robust LM(lag) and Robust LM(error), are further compared. In this case, the Robust LM(lag) is more significant than the Robust LM(error); therefore, the spatial lag model should be selected. This study compared the spatial residuals of the three types of regression models with spatial quantization. It can be seen from the Figure 14 that the residuals of the spatial lag model are much smaller than those of the classical and error models.

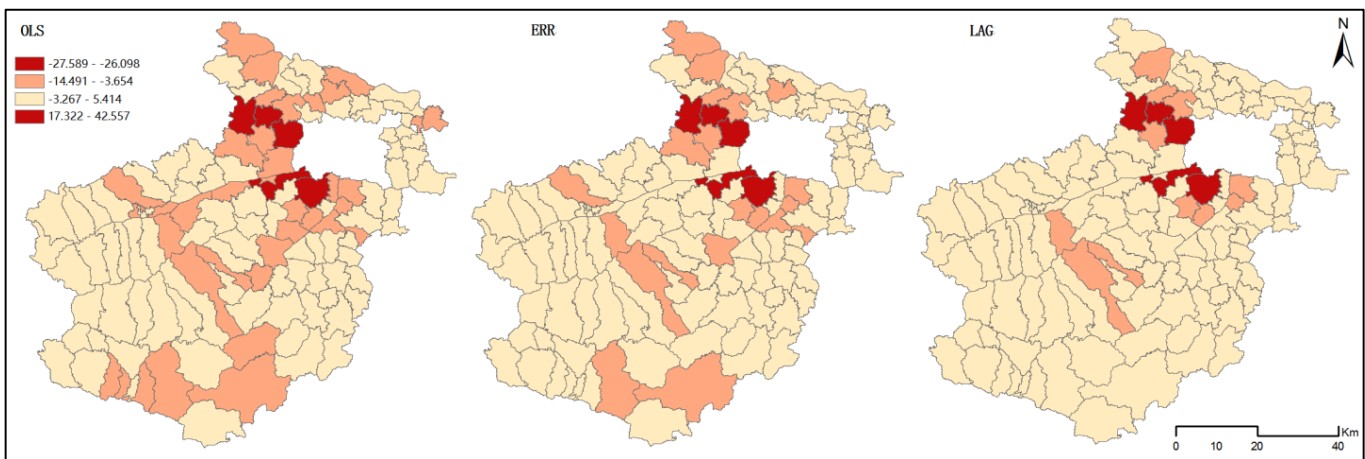

**Figure 14.** Quantification of residual space for three types of regression models.

As shown in Table 5, the results of the spatial lag model indicate that the initial area, slope, elevation, and village farmland had a significant impact on the man-land relationship.

In addition to the distance from the city, other factors were significant (at the 1% significance level), and the calculated results were similar to the real-world scenario.

**Table 5.** Spatial lag model results.

| Impact Factors | Coefficient | Standard Error | t-Statistics | *p*-Value |
|---|---|---|---|---|
| constant | 0.313857 | 0.0883505 | 3.55241 | 0.00038 |
| DEM/m | 0.587033 | 0.070835 | 8.28737 | <0.0001 |
| Slope/° | −0.034160 | 0.000075 | −2.13364 | 0.00387 |
| Rural population/per | −0.031770 | 0.004234 | −7.50271 | <0.0001 |
| Industrial output/million | 4.99E-07 | 6.10E-07 | 0.81721 | 0.00381 |
| Initial village area/Km$^2$ | 6.11E-08 | 2.36E-08 | 2.58325 | 0.00979 |
| Distance from city/Km | 0.00100533 | 0.000746413 | 1.34688 | <0.0001 |
| Cultivated land area/Km$^2$ | −4.83359e-006 | 5.19545e-006 | −0.930349 | 0.17802 |
| Urbanization rate/% | 0.00111366 | 0.000166644 | 6.68283 | <0.0001 |
| Distance from main road/Km | 0.00205353 | 0.00502457 | 0.408698 | 0.068276 |
| Per capita income/yuan | −8.26303e-005 | 4.22326e-005 | −1.95655 | 0.050401 |
| Impact Factors | 2.05259e-007 | 1.74131e-006 | 0.117876 | 0.090617 |

From the absolute value of the regression coefficients, it was observed that slope and elevation have the greatest impact on the coordination of the man-land relationship, followed by the initial village area and cultivated land area. The unbalanced regional development can largely be attributed to the steep slope of the terrain and unfavorable locations far away from the city. The agricultural facilities and living environment are relatively underdeveloped, due to which the rural population is migrating; thus, it appears as though "the people leave, the land stays". The relationship between man and land in these areas tends to be of the type "man decreasing and land increasing". However, with the increasing industrial output value, rural resident population, initial village area, and cultivated land area, the man-land relationship in the region gradually develops towards coordination. This is because of the local economic development that takes place as a consequence of the industrial development of villages and towns, which naturally leads to the return of the rural population and results in the increase of the scale of rural residential land. Since farming is the primary occupation of the majority of rural people, an increase in arable land indicates a comparatively higher degree of development due to agricultural infrastructure projects and a correspondingly higher per capita income and living standard. In such scenarios, the efficiency of intensive utilization of land resources tends to be relatively high. Therefore, the relationship between man and land tends to be of the types "simultaneous increase of people and land" and "decrease of people and land".

## 5. Conclusions

In this study, spatio-temporal big data is used along with 3S technology from the perspective of the human-land relationship to construct an elasticity coefficient model. These methods are combined with the comprehensive index method to further explore the coupling types of rural human-land relations in Luoyang region during 2009–2019 and to explore their spatial divergence patterns in the study area as a whole and in local sample zones. Spatial autocorrelation and spatial autoregressive models are used to reveal the driving mechanisms of the spatio-temporal evolution of rural human-land relationship types. It not only complements the study of the spatio-temporal evolutionary characteristics of human-land relations at the microscopic scale, but also considers spatial correlation in the analysis of the driving mechanisms, and to provide theoretical support for optimal rural land allocation. The research conclusions are as follows:

(1) During 2009–2019, the rural population of Luoyang region continued to decrease, while the scale of rural residential land first increased and then decreased. Per capita rural residential land in Luoyang region reached 225.02 m$^2$/person in 2014, with a trend of serious imbalance in the relationship between people and land. This value

decreased to 180.7 m$^2$/person in 2019, with a slightly lower imbalance. During this period, the main outward flow of rural residential land was arable land, forest and garden land, and other land, while the inward source was mainly arable land, forest, and garden land and grassland. These were focused around the central city.

(2) The type of human-land relationship in rural Luoyang region at the township scale evolves from the dominant imbalance of "people decreasing and land increasing" to the dominant harmonious relationship of "people and land decreasing", thus moderating the conflict between people and land. The villages around the central city show a zonal distribution pattern.

(3) The relationship between rural people and land in Luoyang region is spatially correlated. However, under the combined influence of different environmental and socio-economic factors, the spatial distribution of rural residentials exhibits significant geographical differentiation. The bivariate spatial autocorrelation and the spatial autoregressive model illustrated that the drivers that promote the coordinated development of human-land coupling are, in order of influence, as follows: initial size of rural residentials > arable land area > rural resident population > industrial output value. Similarly, the drivers that cause an imbalance in the type of human-land coupling are, in order of influence: slope > elevation > distance from the city.

(4) With the continuous promotion of the rural revitalization strategy in recent years, most townships in Luoyang region have driven regional economic development through a series of measures, such as developing special industries. These in turn promote the intensive use of land resources. It is evident that, in the process of modernization, the key to regulating the relationship between people and land is the elimination of the drawbacks of the urban-rural dual structured development to achieve more integrated development. In addition, several complex factors influence the spatial and temporal evolution of rural residentials. However, in the process of quantitative analysis, a theoretical perspective is adopted for the treatment of some indicators and the determination of relevant parameters, which introduces a degree of subjectivity into the analysis. Moreover, there is no universal standard for research methods. Therefore, it is necessary to take into account environmental and policy influences and potential uncertainties in future research and to conduct more rigorous quantitative evaluation and analysis of these factors in order to achieve a more reasonable and optimal allocation of land resources.

This paper has made some meaningful research conclusions, but simultaneously there are some problems. In the context of increasingly severe global warming, climate change has also played a significant role in the change of spatial pattern of land resources. However, due to the lack of climate data, systematic research and analysis cannot be done, so climate conditions were not taken into account in this study. In future research, we should consider how to select more influential influencing factors, such as local customs, relevant policies and other factors, so as to further explore the evolution law of the spatial layout of human-land relationship.

**Author Contributions:** Conceptualization, H.W. and Y.Z.; methodology, H.W. and Y.Z.; data curation, H.W. and Y.Z.; validation, Y.Z., W.H. and J.Y.; formal analysis, Y.Z.; resources, H.W. and J.N.; writing—original draft preparation, Y.Z.; writing—review and editing, H.W.; visualization, W.H. and Y.Z.; supervision, J.Y. and J.N.; project administration, H.W. and Y.Z.; funding acquisition, H.W. and W.H. All authors have read and agreed to the published version of the manuscript.

**Funding:** This research was funded by the [Key scientific and technological projects in Henan Province], grant number [212102210105, 222102210046], the [Young Backbone College Teacher Foundation of Henan Province], grant number [2020GGJS130], the [National Sciences Foundation of China], grant number [41771487].

**Institutional Review Board Statement:** Ethical review and approval were waived for this study due to the he project involved in this study was completed with the support of the Henan Provincial Government and a confidentiality agreement was signed, and other institutions were not allowed to have access to the data information.

**Informed Consent Statement:** Informed consent was obtained from all subjects involved in the study.

**Data Availability Statement:** Restrictions apply to the availability of these data. The data sources have been described in detail in Section 2.2.1.

**Conflicts of Interest:** The authors declare no conflict of interest. The funders had no role in the design of the study; in the collection, analyses, or interpretation of data; in the writing of the manuscript; or in the decision to publish the results.

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
