# Peer review of "Spatio-Temporal Evolution and Driving Mechanisms of Rural Residentials from the Perspective of the Human-Land Relationship: A Case Study from Luoyang, China"

_land, doi:10.3390/land11081216_

Round 1
Reviewer 1 Report
Overall, the author did a great job and this research topic is much attractive to the readers which present the driving mechanism of rural residents based on land and human relationship.
The introduction part needs to be more refined adding a few more recent work citations.
The methodology part is well defined and the explanation of the results is perfect.
Where are the industrial output value and initial village area figure number and captions? if it is with figure 2 then the caption should be written below this image or if it is with figure 3 then the frame should be the same.
In lines 125-126, the author mentioned figure 3. spatial quantification and visualization of some drivers but figure no.3 only represents Digital Elevation Model (DEM and Slope which is derived from the DEM! If the industrial image and village area are single images with the DEM then I should advise the author to revise your maps and export them into a single data frame to not confuse the readers.
Page number 8 figure number 6 The sequence is starting from the left side (Spatial distribution of rural settlements transferred to land (09-14) But on page 9 figure number 7 the sequence started from the left side is Spatial distribution of land transferred to rural settlements (14-19). The sequence should be matched. Please order the figures accordingly.
Meanwhile figure 8. should be elaborated with more explanations and scientific justifications. The interpretation of maps is necessary.
I advise the author to please revise your all maps and follow the pattern you have used in Figure 5 with a single Frame.
Author Response
Dear reviewer:
Thank you very much for your suggestions and the reviewers’ insightful comments concerning our manuscript entitled “Spatio-temporal evolution and driving mechanisms of rural residentials from the perspective of the human-land relationship”. Those comments are very valuable and helpful for improving the quality and readability of our paper, as well as the important guiding significance to our future researches. We have revised the manuscript, and would like to re-submit it for your consideration. We have addressed the comments raised by the reviewers, and revised the questions one by one. Point by point responses to the reviewers’ comments are listed below this letter.

Reviewer 2 Report
To me the paper can be published like this. I won't add anything to this text
Author Response
Dear Editor:
Thank you for your recognition of this study! Your approval of this study is the greatest encouragement and motivation for us.
We are very grateful for your and reviewers’ warm work earnestly.

Reviewer 3 Report
The detailed comments are as follows:
The research objective in the abstract is blurred.
Taking into account the title and the empirical part, a question arises as to their coherence (the work concerns only a selected unit in China, which the title does not suggest).
It is incomprehensible that the study covers the city (Luoyang City), and the study concerns settlements and rural population. In my opinion, from the point of view of the international reader, it is necessary to explain the essence of the administrative division in China and the definition of the concept of ruralness, which I hope will resolve the indicated doubt.
The manuscript describes the study area (2.1), but the issues related to the specificity of the rural population (e.g. number of rural population) are omitted, which is unsatisfactory given the purpose of the study. There is no justification as to why Luoyang City was selected for the research.
A weak point of the work is the lack of a general discussion of the obtained results with the conclusions of other authors and a description of the limitations of the research undertaken.
Descriptions of the graphic elements require correction (e.g. no title of Table 1, unclear title of Fig. 13; illegible diagram 16, it is not always known what the entry 09-14, 14-19 means).
The work requires intensive improvement of the text editing, which also applies to the references.
Author Response

(The authors gave the same response as above.)
